**PLOS** NEGLECTED TROPICAL DISEASES

# Independent evaluation of *Wolbachia* infected male mosquito releases for control of *Aedes aegypti* in Harris County, Texas, using a Bayesian abundance estimator

**Saul Lozano**[1]*, **Kevin Pritts**[2], **Dagne Duguma**[3¤], **Chris Fredregill**[3], **Roxanne Connelly**[1]

**1** National Center for Emerging and Zoonotic Infectious Diseases, Division of Vector-Borne Diseases, Centers for Disease Control and Prevention, Fort Collins, Colorado, United States of America, **2** Western Gulf Center of Excellence for Vector-Borne Diseases, University of Texas Medical Branch, Galveston, Texas, United States of America, **3** Harris County Public Health, Mosquito and Vector Control Division, Houston, Texas, United States of America

¤ Current address: Broward County Mosquito Control, Pembroke Pines, Florida, United States of America
* nkq3@cdc.gov

**Data Availability Statement:** All relevant data are within the manuscript and its Supporting Information files.

## Abstract

Among disease vectors, *Aedes aegypti* (L.) (Diptera: Culicidae) is one of the most insidious species in the world. The disease burden created by this species has dramatically increased in the past 50 years, and during this time countries have relied on pesticides for control and prevention of viruses borne by *Ae. aegypti*. The small number of available insecticides with different modes of action had led to increases in insecticide resistance, thus, strategies, like the "Incompatible Insect Technique" using *Wolbachia*'s cytoplasmic incompatibility are desirable.

We evaluated the effect of releases of *Wolbachia* infected *Ae. aegypti* males on populations of wild *Ae. aegypti* in the metropolitan area of Houston, TX. Releases were conducted by the company MosquitoMate, Inc. To estimate mosquito population reduction, we used a mosquito abundance Bayesian hierarchical estimator that accounted for inefficient trapping. MosquitoMate previously reported a reduction of 78% for an intervention conducted in Miami, FL. In this experiment we found a reduction of 93% with 95% credibility intervals of 86% and 96% after six weeks of continual releases. A similar result was reported by Verily Life Sciences, 96% [94%, 97%], in releases made in Fresno, CA.

## Author summary

*Aedes aegypti* is one of the most important mosquito species because females can potentially carry pathogens that cause disease. These diseases have a tremendous impact worldwide making this species an important target of control.

We evaluated a mosquito control strategy independently of the company that developed the method while the company tested it in Harris County, TX. The strategy relies on

**Funding:** The mosquito releases were supported by the CDC Hurricane Cooperative Agreement Funding through the Texas Department of State Health Services contract to Harris County Public Health (HHS000371500029, MosquitoMate, Inc. conducted the releases of WIMs as the sub-contracting agency) and the Cooperative Agreement (U01CK000512, KP) also funded by the Centers for Disease Control and Prevention. SL, DG, CF, and RC did not receive specific funding for this work. The funders had no role in study design, data collection and analysis, decision to publish, or preparation of the manuscript.

**Competing interests:** The authors have declared that no competing interests exist.

a bacterium (*Wolbachia* sp) that causes changes in the sperm of infected males, preventing uninfected female mosquitoes from producing viable eggs (phenomenon known as cytoplasmic incompatibility). *Wolbachia*-infected males are released in large numbers (inundative releases) to outcompete wild-type *Wolbachia*-free males and reduce the population of existing *Ae. aegypti* mosquitoes.

We observed a sustained reductions > 90% in the number of females very likely because of the intervention conducted in Harris County, TX. The results we observed were very similar to observations made by others in Miami, FL and in Fresno, CA. However, more experiments (following randomized cluster designs) should be performed to increase the statistical power while controlling for environmental factors that could contribute to fluctuations in mosquito populations and trapping variations.

## Introduction

Among disease vectors, *Aedes aegypti* (L.) (Diptera: Culicidae) is one of the most insidious species in the world. It feeds almost exclusively on humans and, accordingly, it is markedly well adapted to live in the human environment. In the past 50 years the disease burden created by *Ae. aegypti* has increased considerably [1] despite reductions in the number of dengue cases in the middle of the 20[th] century [2]. In the United States from 2015–2017, local populations of *Ae. aegypti* infected humans with Zika virus in Florida, Texas, Puerto Rico, and the US Virgin Islands [3]. Among the viruses transmitted by *Ae. aegypti*, dengue virus (DENV) has prominence, with an estimated 400 million cases per year [4,5]. Given this staggering impact, countries have placed great emphasis on the control of dengue; disease control is mainly carried out using insecticides to reduce the vector population.

Dependence on insecticides has placed tremendous selective pressure on *Ae. aegypti*, and consequently insecticide resistance has increased rapidly [6]. At the same time, the scarcity of active ingredients approved for public health protection with varied modes of action have made insecticide resistance a global issue [7,8]. As such, novel mosquito control strategies are desirable. One such approach is the sterilizing effects of *Wolbachia* bacteria by means of cytoplasmic incompatibility [9–12]. Mosquito control using the sterility obtained via cytoplasmic incompatibility has been termed "Incompatible Insect Technique" (IIT) [9] to differentiate it from the "Sterile Insect Technique" (SIT) [13] that uses radiation (or a chemical) for sterilization. Two companies, MosquitoMate, Inc. (Lexington, KY) and Verily (subsidiary of Alphabet, Inc., Mountain View, California) have reached the implementation phase of IIT for the local suppression of *Ae. aegypti* populations [14–16], and recently reported localized reductions of 78% in Miami, FL [14] and 96% Fresno, CA [16].

In 2017, Harris County Public Health Mosquito and Vector Control Division (HCPH MVCD) received CDC Hurricane Cooperative Agreement Funding through the Texas Department of State Health Services contract to implement and evaluate traditional as well as novel mosquito vector control approaches in Harris County, TX. Harris County is the third most populous county in the US [17] (4,713,325 residents) and includes the City of Houston which is the fourth most populous city [17] (2,320,268 residents) in the US. The goal of the funding was to increase the vector control capacity of HCPH MVCD to better respond to increased vector-borne disease risk in the region. *Aedes aegypti* and *Aedes albopictus* (Skuse) (Diptera: Culicidae), two important vectors of Dengue and other emerging arboviral diseases, have co-occurred in Harris County for 34 years.

One of the novel approaches evaluated in Harris County includes the suitability of an auto-cidal approach using releases of *Wolbachia*-infected males (WIM). The goal of the project reported here was to independently evaluate the efficacy of the WIM releases and examine the effects on the abundance of local *Ae. aegypti* populations. In addition to tracking the abundance of *Ae. aegypti*, we tracked *Ae. albopictus*, which is commonly found in Houston. Here we present an independent evaluation of WIM releases for the suppression of *Ae. aegypti*. Additionally, we describe, and test, the abundance estimation approach we used (N-Mixture Bayesian hierarchical model [18]); this approach is relatively novel in mosquito ecology, being used previously to estimate the abundance of *Ae. albopictus* [19], and *Ae. aegypti*, but with a Mark Release Recapture Component [20] (MRR).

## Methods

### Rearing, sex separation, delivery, and releases of *Wolbachia* infected males

All decisions and activities related to mosquito rearing, infection with the *Wolbachia pipientis* wAlbB strain, mosquito separation by sex, and the inundative application of WIM, were solely those of MosquitoMate, Inc (MM). WIM were released at several points inside the treatment area three times a week during the June 17th–August 28th, 2019 period (Fig 1.), following the requirements set in an Environmental Protection Agency experimental user permit (EUP) [21]. The week numbers corresponds to the International Organization for Standardization (ISO) definition of week number [22].

### Mosquito population surveillance

Two areas in Harris County, TX within Houston's metropolitan area, previously selected by MM, served as an untreated area (Fig 2: UA), and a treatment area (Fig 2: TA). The areas were 20.3 km. apart; female mosquitoes tend to stay close to their eclosion site but have been found to travel up to ~600 meters in MRR experiments [23,24]. No mosquito control was conducted by HCPH MVCD in either area during the study period, however, mosquito control conducted by the residents was not precluded or recorded.

During household recruitment, we asked for permission to place a trap on the property and enter the property to service the trap for the duration of the surveillance. No compensation was offered, or given, for the use of the properties. We recruited 27, and 28 households in the UA, and the TA, respectively; households monitored by MM staff were excluded from our recruitment effort. We placed a single BG-Sentinel 2 trap (Biogents AG, Germany) at each participating household, mainly in front yards next to windows and doors under the cover of vegetation when available. Following the same trap configuration as MM, we baited the traps with one long lasting BG-Lure (Biogents AG, Germany), and dry ice (2 kg) in a cooler with a top nozzle. The traps ran continuously for 48 hrs. with a change of collection bag and dry ice at 24 hrs., effectively trapping twice per week on subsequent days. Specimens of *Ae. aegypti* and *Ae. albopictus* were processed (sexed, counted, and keyed [25]) individually for each day and trap; all males and other species were discarded. MM started its inundative releases on week 25, and our collections began during week 28. Our sampling was conducted for eleven weeks from week 28 through week 38. *Aedes albopictus* was not the intended target of the WIM intervention at our study site, but since *Ae. aegypti* and *Ae. albopictus* are sympatric there was interest in knowing whether decreasing the abundance of *Ae. aegypti* would increase the abundance of *Ae. albopictus*.

To ensure trapping uniformity, we recorded if the trapping was successful (i.e., traps were operating correctly, traps were undisturbed, etc.), we also recorded the trapping time from trap setup to collection bag retrieval. Traps that statistically deviated from the mean trapping

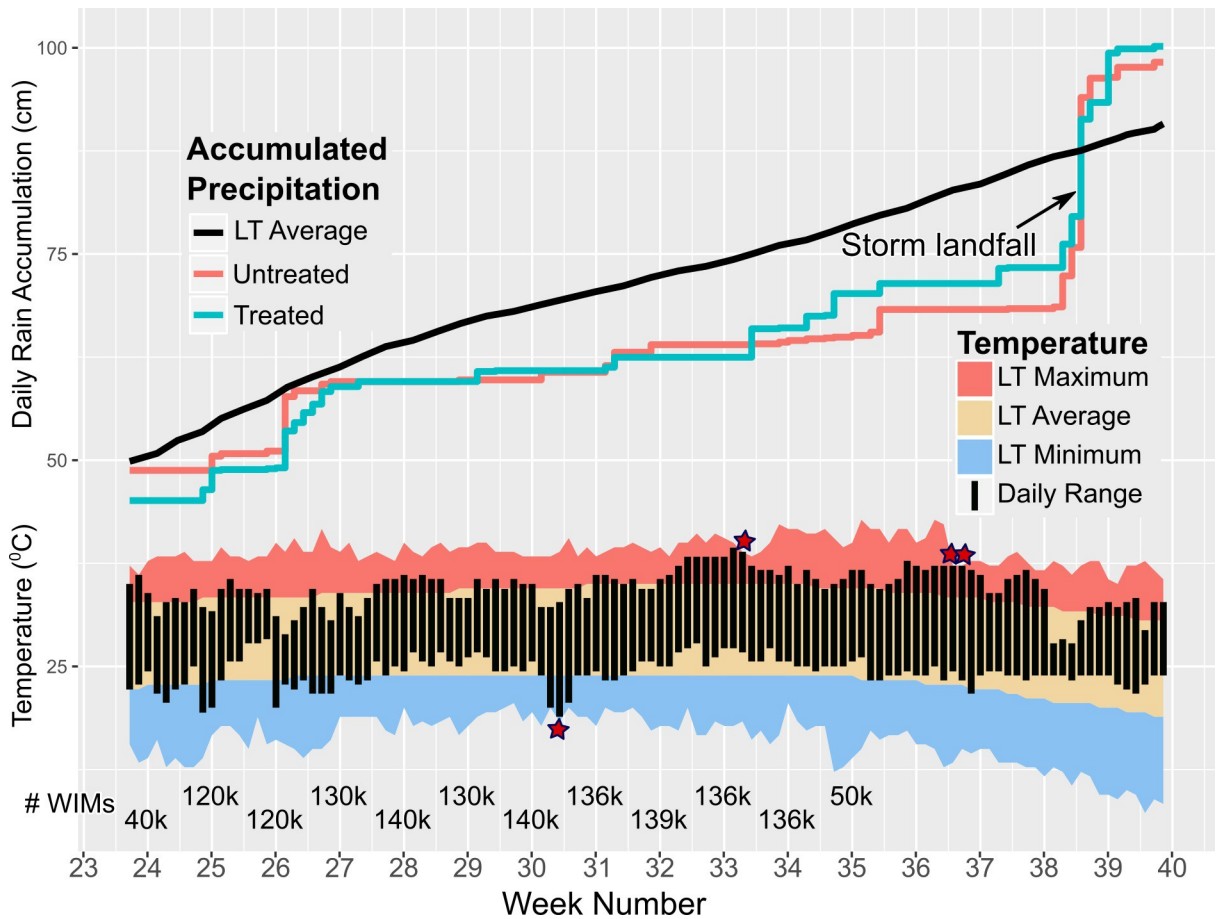

**Fig 1. Average and observed daily cumulative rainfall, and temperature.** Rainfall for each area was recorded at the nearest Harris County Flood Warning System weather station. The Long-term Rainfall (LT rainfall) is a 20-year average (1981–2001) recorded at Houston's George Bush Intercontinental Airport (IAH). The LT Maximum, Average, and Minimum are 20-year averages (1981–2001) recorded at IAH. The observed daily temperature (bar; top of the bar observed maximum; bottom of the bar observed minimum) was recorded at IAH. Red stars show days when the temperature broke or tied the LT maximum or minimum. The number of released WIMs per week is presented at the bottom of the plot (k = x 1000). Climate and Weather data Source: National Weather Service and Harris County Flood Warning System. Raw data in S3 File.

time were not included in the analysis. Time trapping differences were evaluated by fitting a Student's *t* distribution [26] to the trapping time. Differences in mean trapping time were evaluated using 95% credibility intervals (CI) where the most probable (or expected) value was the 50th percentile (pct), the lower limit the 2.5th pct, and the upper limit was the 97.5th pct of the posterior distribution [27]. In similar fashion to confidence intervals, the most probable estimates are presented before the limits and the limits are presented inside parenthesis, i.e., 50th pct (2.5th pct, 97.5th pct). Non-overlapping CIs were considered statistically different with a 95% probability [28].

To assist in the evaluation of population changes, we also obtained publicly available meteorological data (Fig 1.) The daily accumulated rainfall for the UA and the TA was recorded at the nearest Harris County Flood Warning System [29] weather station (Fig 2.) The "Normal Cumulative Precipitation" [30], a 20-year average (1981–2001), was recorded at Houston's George Bush Intercontinental Airport (IAH). The daily temperatures [31], as well as the temperature "normals" [30], were recorded at IAH.

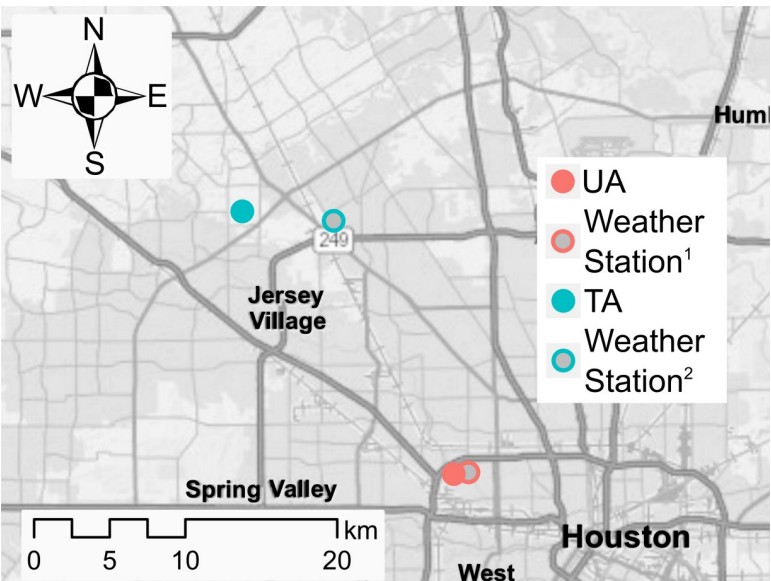

**Fig 2. Location of the untreated and treated areas in Harris County, TX, and the closest weather station operated by Harris County Flood Control.** Only precipitation data was available from these stations. Map generated with ArcMap [51] contains information from OpenStreetMap and OpenStreetMap Foundation, which is made available under the Open Database License.

## Abundance estimation: The N-mixture model

We used an N-Mixture Bayesian hierarchical model [18] that takes into account poor detection due to low trapping efficacy. This approach has been previously used in mosquito ecology and has potential applications in analyzing mosquito surveillance data and guide control actions. It was used to estimate the abundance of *Ae. albopictus* [19], and *Ae. aegypti* in a Mark Release Recapture (MRR) study [20].

The N-Mixture model addresses the issue of finding the mosquito abundance, despite the distortion created by the trapping, by assuming that the number of collected mosquitoes is the result of two processes:

$$\text{a trapping process: } C_{ijk}|N_{ik} \sim \text{Binomial}(\tau_{ik}, N_{ik}), \tag{1}$$

$$\text{and a population state process: } N_{ik} \sim \text{Poisson}(\lambda_{ik}). \tag{2}$$

$$\log(\lambda_{ik}) = \beta_{0_{ik}} + \beta_{1_{ik}} \cdot treatment_{ik} + \beta_{2_{ik}} \cdot time_{ik} \tag{3}$$

The trapping process produces the number of mosquitoes caught (*C*) in trap *i* in trapping occasion *j*, in week *k*, and it was model with a Binomial distribution. Notice that each trapping occasion is a "repeated-measure" of the population. The $\tau_i$ parameter represents the trapping efficacy of each trap, or the probability that trap *i* caught all the mosquitoes in its area of influence (e.g., a value of 1.0 would denote a trap that caught 100% of the mosquitoes). The second parameter, $N_{ik}$, a latent variable, represents the number of mosquitoes that was present at each trapping location.

The population process produces the number of mosquitoes at each site ($N_{ik}$) and in accordance with ecological theory, it was assumed to follow a Poisson distribution [32], the only

parameter $\lambda$ was model as a linear regression with the *treatment* (UA = 0, TA = 1) and the *week* (1–11) as covariates. Notice that in itself, the population process can be considered a Poisson linear model (a.k.a. Poisson regression). In this context the $\lambda$ parameter represents the mean number of females in the trapping area, $\beta_1$ the effect of the treatment on $\lambda$, and $\beta_2$ the effect of time on $\lambda$; values for $\beta_1$, or $\beta_2$ statistically different from zero denote that the covariate had a statistical influence on the mean number of females with a probability $\geq$ 95%. We directly measured spatial aggregation in a separate model; the description and the results are presented in the S1 File.

## Evaluating the N-mixture approach

We evaluated the N-mixture approach by comparing the fitted models to the raw data. The models' fit to the data was assessed visually using "Post predictive checks" (PPCs) [27,33]. In addition to PPCs, we tested the N-mixture approach numerically by evaluating the predictions of the total number of females against a known number of released *Ae. aegypti* females; the total number of females was estimated as $\Sigma N_i$. For this challenge we used published and publicly available *Ae. aegypti* females MMR data from Rio de Janeiro [20]. The releases were divided in two MRR experiments: the first experiment had four groups of females marked blue ($N = 500$), pink ($N = 500$), yellow($N = 500$), or green($N = 500$), and the second experiment had a single release of 2000 blue females. A prediction was considered correct if the number of released females fell inside the estimated 95% CI.

The parameters for the N-Mixture model, and the Student's *t*, were estimated using Bayesian inference with the help of JAGS version 4.2.0 [34], the jagsUI library [35], and the R language [36]. A working example of the N-Mixture model (R language code) is in S2 File.

## Reductions in abundance

The reductions were estimated only when the abundance between the areas, in the same week, were statistically different. The most probable reduction was estimated using the $50^{th}$ pct, i.e., $(1 - (TA_{50th\ pct} / UA_{50th\ pct})) \times 100$. To contrast the reductions, we used a conservative approach, the lower limit was estimated using the smallest difference between CIs $(1 - (TA_{97.5th\ pct} / UA_{2.5th\ pct})) \times 100$; the upper limit was estimated comparing the largest possible distance between the CIs $(1 - (TA_{2.5th\ pct} / UA_{97.5th\ pct})) \times 100$.

## Results

After trap placement, we estimated the trapping area by calculating the minimum convex polygon (a.k.a. Convex Hull [37]) plus a 50-meter buffer around the traps. The UA had an area of 29.6 hectares (ha) while the TA had an area of 18.6 ha. To make the abundance estimation comparable between areas, and not too close to zero, the abundance estimations were divided by one-tenth of a hectare or 29.6/10 and 18.6/10 ha (ha/10 = $ha^{-0.1}$), respectively.

We discovered during data validation that the trapping time of the first trapping (first day of week 28), in both areas, was statistically different from other weeks. The mean trapping time during the first trapping was 27 (26, 27) hrs., and 18 (18, 19) hrs. in the UA, and TA respectively. In comparison, the mean trapping time on other days was 23 (23, 24) hrs. Given the statistical differences (i.e., the credibility intervals did not overlap), the first trapping day was removed from subsequent analysis. Coincidentally, the second trapping of week 38 was not conducted due to hazardous conditions created by the landfall of Tropical Storm Imelda in Houston (Fig 1).

Altogether, we conducted 558 successful trapping events in the UA, and 551 in the TA; of 101 unsuccessful trapping (no CO2, battery disconnected, missing collection bag, etc.), 55

were due to weather. We collected a total of 5,752 *Ae. aegypti* females and 5,926 *Ae. albopictus* females in the UA, while in the TA we collected 904 *Ae. aegypti* females and 4,932 *Ae. albopictus* females. Missing data points from unsuccessful trapping were passed to the inference program as data not available ("NA" in the R language), that is, the data was not adjusted to account for the missing trappings.

## N-Mixture predictive power

From the first release in Rio de Janeiro, only 67 blue females were recovered in 20 traps; the N-mixture approach predicted $N = 461$ (138, 1559) blue females would be in the trapping area. Only 52 pink females were recovered in 21 traps; we predicted $N = 432$ (147, 1194) females. Only 35 yellow females were recovered in 16 traps; we predicted $N = 454$ (58, 2466) females. Only 30 green females were recovered in 10 traps; for this color it was not possibly to predict the number of females because the Bayesian chains did not converge (i.e., the fitting algorithm did not find a proper solution for the model's parameters), likely the result of the small number of recaptured females (6%), the lower number of positive traps, or the combination. For the second release (2000 blue marked females) the N-mixture approach predicted $N = 1715$ (645, 3629). Given the results from the first and second release, we can say that the N-Mixture approach can accurately predict the number of females in an area.

## *Aedes aegypti* abundance in Harris CO., TX

We evaluated the fit of the Houston N-Mixture models using PCCs. We observed that the models appropriately described the trapping data for *Ae. aegypti* (and *Ae. albopictus*) in every week. (Fig 3 and S1 Fig has PPCs for both areas) because there is large agreement between the fitted models (blue) and the raw data (red) histograms.

In Table 1 we present the regression results describing the effect of the treatment and time on the mean number of females ($\lambda$). On weeks 28, 29, and 30, the intercept was greater than zero, with a value of ~ 3.5, in the remaining weeks the intercept remaining statistically zero. The treatment's effect was statistically different from zero every week, demonstrating that the treatment influenced the mean number of females, whereas time (in weeks) did not.

The abundance of *Ae. aegypti* in the UA (Fig 4: red)—expressed as the mean number of *Ae. aegypti* females per trap ha$^{-0.1}$ ($\lambda$)—remained stable from week 28 through week 36, with non-statistical increases on week 33 and 35, as indicated by the red dotted lines marking the 95% CI of week 28. By week 36, $\lambda$ returned to values equal to previous weeks, but in weeks 37 and 38 there was a statistically significant decline in relation to week 28.

In contrast, $\lambda$ in the TA (Fig 4: blue) showed a steady decline from week 28 through week 31. The $\lambda$ estimates were statistically different from each other in weeks 28, 29, 30, and 31. After week 31, the *Ae. aegypti* population did not recover and remained below two females per trap ha$^{-0.1}$, as shown by the blue lines that mark 95% CI for week 31 ($\lambda = 1.8$ (1.2, 1.8)).

The treatment's measurable effect, along with the stable mosquito population in the UA, offered compelling proof that the observed decreases were the result of the WIM releases. Though there were no statistical differences in week 28 between the UA and the TA, by week 29 the reduction was statistically different (UA $\lambda = 12.5$ (9.9, 16.5); TA $\lambda = 6.7$ (5.0, 9.2)), representing a reduction of 47%. By week 30, the reduction was 82% (UA $\lambda = 14.8$ (11.6, 19.5); TA $\lambda = 2.7$ (1.9, 3.7), and from week 31 up to week 37, the reductions were ~94% (e.g., week 31, UA $\lambda = 13.4$ (10.3, 18.6); TA $\lambda = 1.1$ (0.7, 1.7)). The highest reductions started six weeks after the start of the releases. Week 38, and to some extent week 37, are hard to interpret due to the statistically significant reductions in $\lambda$ in the UA.

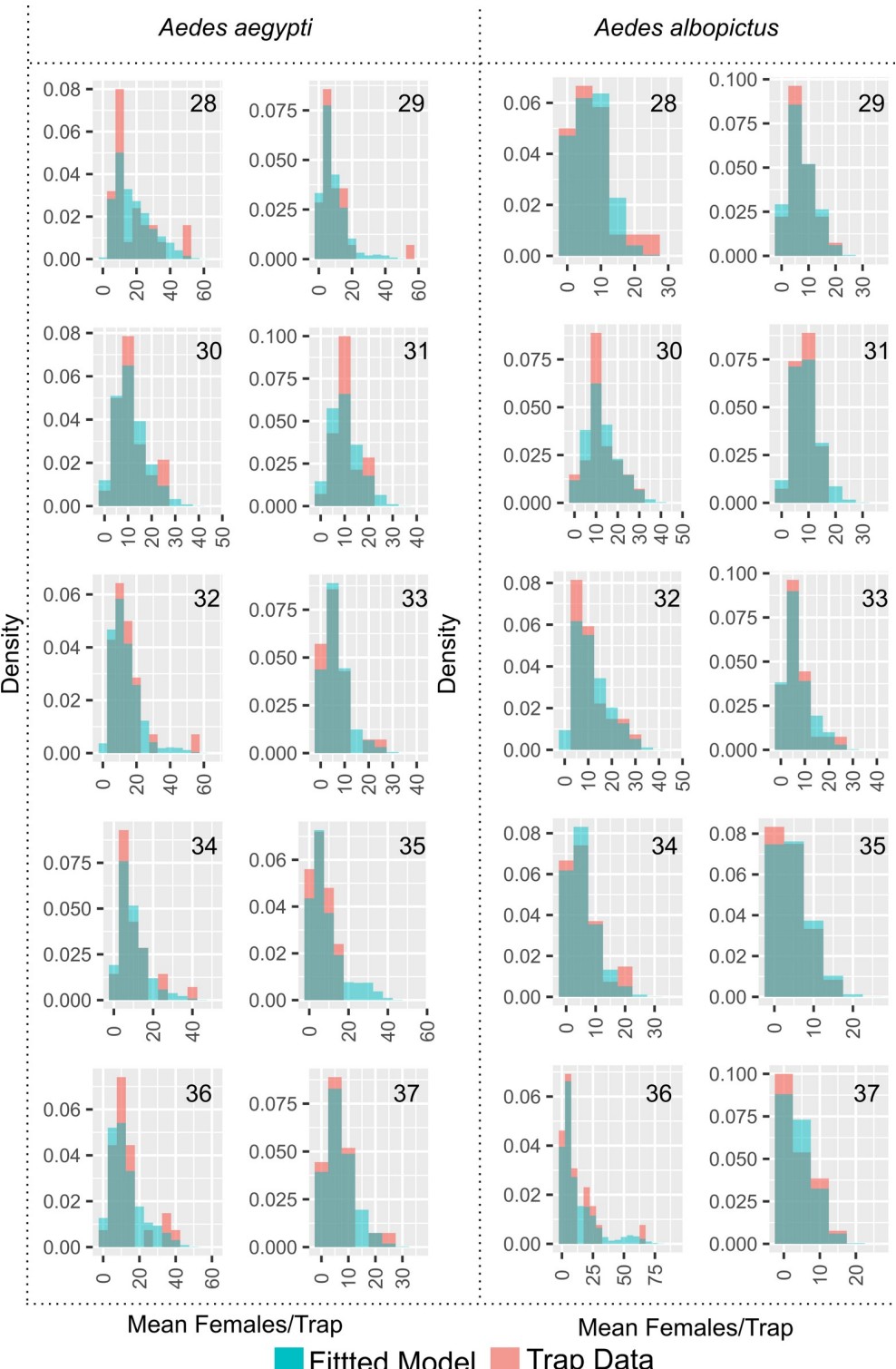

**Fig 3. N-mixture fitted models for *Ae. aegypti* and *Ae. albopictus* from the untreated area by week.** Blue histograms are the best N-Mixture fitted models; Red histograms are the raw trapping data (bin width = 5); the number represent the sampling week. A properly fitted model will cover most of the trapping data histogram. For example, in week 28 the model underestimated the number of traps in the 10–14 and 40–44 *Ae. aegypti* category (the red bars are larger than the blue bars).

**Table 1.** *Aedes aegypti* Regression Table Results by Week.

| Week | Intercept | Treatment | Time |
|------|-----------|-----------|------|
| 28 | 3.7 (2.6, 4.5)* | -0.3 (-0.6, -0.02)* | 0.1 (-0.7, 1.1) |
| 29 | 3.4 (1.2, 4.9)* | -1.2 (-1.6, -0.8)* | 0.1 (-0.6, 1.2) |
| 30 | 3.4 (0.3, 5.4)* | -2.3 (-2.7, -1.9)* | 0.1 (-0.5, 1.2) |
| 31 | 3.1 (-0.6, 5.6) | -3.1 (-3.6, -2.6)* | 0.1 (-0.5, 1.1) |
| 32 | 3.1 (-1.2, 5.9) | -3.2 (-3.7, -2.7)* | 0.2 (-0.4, 1.0) |
| 33 | 3.2 (-1.6, 6.3) | -5.1 (-6.0, -4.3)* | 0.2 (-0.4, 1.0) |
| 34 | 2.6 (-2.3, 6.1) | -3.5 (-4.1, -2.9)* | 0.1 (-0.4, 1.0) |
| 35 | 2.8 (-2.6, 6.5) | -4.4 (-5.1, -3.7)* | 0.2 (-0.3, 1.0) |
| 36 | 2.4 (-3.0, 6.5) | -4.2 (-4.9, -3.6)* | 0.2 (-0.3, 1.0) |
| 37 | 2.0 (-3.4, 6.2) | -2.7 (-3.2, -2.1)* | 0.1 (-0.3, 1.0) |
| 38 | 0.8 (-4.4, 5.4) | -1.7 (-2.5, -1.0)* | 0.1 (-0.4, 1.0) |

* value is statistically different from zero with a 95% probability; the expected value is the 50th percentile of the parameter's posterior distribution, the CIs were drawn using the 2.5th and 97.5th percentiles of the parameter's posterior distribution.

### *Aedes albopictus* abundance

We demonstrated in Fig 3 and S1 Fig that the fitted model for this species accurately described the trap data in every week, and we present the regression results for *Ae. albopictus* in Table 2. Only weeks 28, 29, and 30 had an intercept greater than zero, with a value of ~ 3. In terms of the treatment effect, in comparison to *Ae. aegypti*, the treatment had a minor impact on the mean number of *Ae. albopictus* females, with most values less than one but statistically greater than zero; week 33 showed the largest effects. This indicates that the presence of the treatment increased the mean number of *Ae. albopictus* females, likely due to the decrease in the number of *Ae. aegypti*. In terms of the effect of time, this covariable was not different from zero in any of the weeks, and thus time had no impact on the mean number of females.

The $\lambda$ (females per trap ha$^{-0.1}$) for *Aedes albopictus* in the UA (Fig 5: red) was more variable than *Ae. aegypti*, however, given the uncertainty around $\lambda$, the differences were not statistically significant in most weeks as demonstrated by the overlapping CIs drawn from week 29 ($\lambda$ = 10.3 (7.9, 13.9). Weeks 28 ($\lambda$ = 5.3 (4.2, 7.1)), 36 ($\lambda$ = 39.5 (29.4, 52.7)), and 38 ($\lambda$ = 1.0 (0.7, 1.5)) are clearly statistically different from the other weeks and from one another. The $\lambda$ at the TA showed a similar pattern to the UA, but with higher values (Fig 5: blue; week 29 $\lambda$ = 39.5 (29.4, 52.7)), and only week 38 ($\lambda$ = 2.8 (2.0, 3.8)) was statistically different from the other weeks.

### Discussion

In order to understand experimental results and reach the right conclusions, proper data analysis is required. Without a doubt, statistically reliable estimates of population abundance are necessary for evaluating a vector control intervention. However, because females are particularly well-suited to finding people, which is their primary source of blood, it is difficult to estimate the number of *Ae. aegypti* in a given region. Due to host-seeking adaptations in the female, trap performance is poor when people are around, resulting in an excess of traps with low mosquito counts and a dearth of traps with "high" numbers [18,32]. For these reasons, we used the N-mixture model.

We demonstrated that the N-mixture model, cannot only describe the trapping data appropriately, but also predict a known number of released female mosquitoes in an area. However, we observed an underestimation of the true number of marked females (8–14%), for the MMR

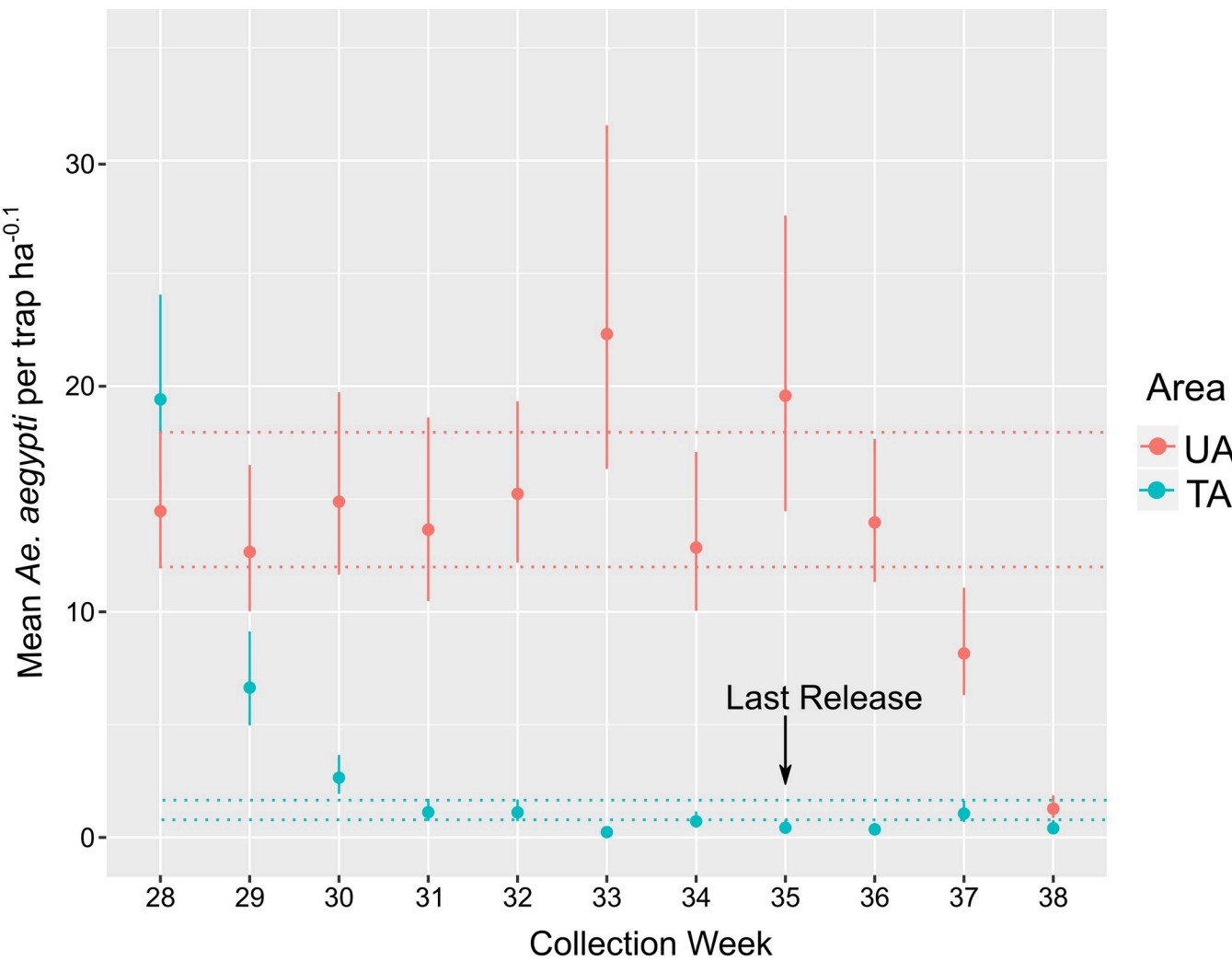

**Fig 4. Relative abundance of *Ae. aegypti* females per week in the untreated and treated area.** The center dot denotes the most probable value. Error bars denote 95% credibility intervals (see the "Statistical Methods for Abundance Estimation" section for description of the mean and credibility intervals). Red dotted line represents the credibility interval for the UA abundance in week 28. Blue dotted line represents the credibility interval for the TA abundance in week 31. If the credibility intervals do not overlap the means are considered statistically different with a probability of 95%.

experiments. Given the lack of publicly available MRR data, we cannot establish that the observed underestimation is a common feature of N-mixture models. Contextualizing the possible underestimation impact on our study, assuming that underestimation is present in all N-mixture models, the differences in *Ae. aegypti* abundance between the UA and TA six weeks post-intervention are considerably larger than 14%. Additionally, the underestimation would be present in the estimates of both areas.

We also observed that the CIs upper limits for the MRR predictions were large (~2–5 times the true value); larger intervals appear to be related to lower recapture numbers and the effect of the distance from the released site [20]. We addressed this issue by only calculating an abundance reduction when weekly estimates were statistically different and by comparing the upper and lower limits of the abundance and not only the most probable values, i.e., if measurements for a particular week had large CIs, the reduction's uncertainty will also grow considerably.

**Table 2.** *Aedes albopictus* **Regression Table Results by Week.**

| Week | Intercept | Treatment | Time |
|---|---|---|---|
| 28 | 2.7 (1.5, 3.5)* | 0.8 (0.4, 1.1)* | 0.1 (-0.7, 1.2) |
| 29 | 3.2 (0.9, 4.6)* | 0.5 (0.2, 0.9)* | 0.1 (-0.6, 1.3) |
| 30 | 3.5 (0.2, 5.5)* | 0.4 (0.1, 0.8)* | 0.2 (-0.5, 1.3) |
| 31 | 2.8 (-0.8, 5.2) | 0.1 (-0.3, 0.5) | 0.2 (-0.4, 1.0) |
| 32 | 2.8 (-1.3, 5.5) | 0.6 (0.2, 0.9)* | 0.2 (-0.4, 1.0) |
| 33 | 3.0 (-1.6, 6.1) | 0.5 (0.0, 1.0)* | 0.2 (-0.3, 0.9) |
| 34 | 2.2 (-2.3, 5.6) | 0.8 (0.3, 1.2)* | 0.1 (-0.3, 0.8) |
| 35 | 1.9 (-2.8, 5.6) | 1.7 (1.2, 2.1)* | 0.1 (-0.3, 0.7) |
| 36 | 2.9 (-2.4, 6.8) | -0.4 (-0.8, 0.0) | 0.2 (-0.2, 0.8) |
| 37 | 1.7 (-3.3, 5.8) | 0.6 (0.1, 1.0)* | 0.1 (-0.3, 0.6) |
| 38 | 0.6 (-4.1, 4.9) | 0.4 (-0.1, 0.9) | 0.0 (-0.4, 0.5) |

* value is statistically different from zero with a 95% probability; the expected value is the 50th percentile of the parameter's posterior distribution, the CIs were drawn using the 2.5th and 97.5th percentiles of the parameter's posterior distribution.

The N-mixture estimations could be greatly improved by adding relevant covariates that affect the mosquito population processes (rain accumulation, rainfall event lag, temperature,

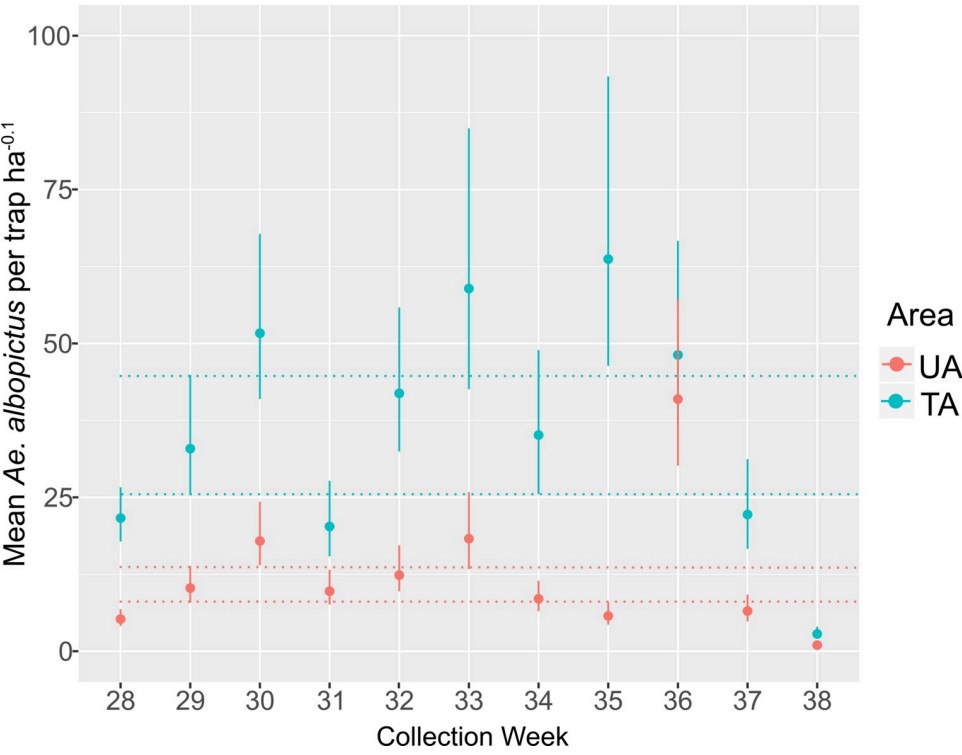

**Fig 5. Relative abundance of *Ae. albopictus* females per week in the untreated and treated area.** The center dot denotes the most probable value of the estimate. Error bars denote 95% credibility intervals (see the "Statistical Methods for Abundance Estimation" section for description of the mean and credibility intervals). The red dotted line represents the credibility interval for the UA abundance in week 29. The blue dotted line represents the credibility interval for the TA abundance in week 29. If the credibility intervals do not overlap the means are considered statistically different with a probability of 95%.

etc.), the trapping processes (wind speed, time of day, proximity to people, age of the lure, etc.), and adding a dispersion process (housing density, people density, distribution of larval sites, attraction to households, etc.). There are clear advantages in the estimation of the parameter $\lambda$, which grants access to a large compendium of ecological theory and it is a baseline for comparison to other spatial patterns [32]. In addition to the incorporation of space, time could also be incorporated in the form of a Poisson inhomogeneous process [38, 39]. Using the N-mixture approach necessitates an appropriate trapping strategy, which includes careful trap placement [40,41], in order to reduce the uncertainty around the trapping efficacy estimate [19], which may interfere with vector control program logistics. However, any vector control program would benefit greatly from having an accurate, unbiased estimate of population reductions.

The regulatory restrictions outlined in the experimental use permit (United States Environmental Protection Agency, 89668-EUP-3 [21]) and MosquitoMate contractual responsibilities, which restricted the entire area of application, resulted in limitations in our experimental design. As a result of these constraints, there was only one untreated and one treated area. A better design would have been to switch treatment from the TA to the UA after achieving the desired level of suppression and then wait for the *Ae. aegypti* population to recover. It is unclear, however, how the weather would have affected the recovery.

Water is so essential in the life cycle of mosquitoes that it could be argued that the observed *Ae. aegypti* reductions in the TA were the result of reduced precipitation, and not of the WIM treatment. However, the TA received more rain than the UA, therefore, it is safe to assume that the reductions observed in the TA were mostly the result of the WIM control intervention.

In a recent WIM intervention in Miami, FL [14] the authors estimated the female abundance as the arithmetic mean of all the traps in a monthly collection. To compare the abundance between the untreated area and the treated area, the authors used the non-parametric Wilcoxon rank-sum (*Ws*) test on each pair of monthly collections. Keeping in mind the analysis differences, the intervention in Florida had a smaller reduction in the mean number of females (78%) than the one observed in Harris CO., TX (92%). We observed reductions of 82% in the mean number of females five weeks after the start of the releases, and after that, the reductions were about 92%. Furthermore, unlike in Florida, the Houston results were obtained without dividing the TA into "Edge" and "Center," which appears to contradict the observation that reductions are greater in a treated area's center. However, the reductions observed in our study would agree with the observation made in Florida about WIM dispersal, indicating that the control was effective at both the edges and the central section, and that the migration rate in Houston, TX was lower than in Miami, FL. The reduced migration could be the result of high temperatures [42] or reduced movement; the Houston study areas were located in highly urbanized zones [43].

Another recent control intervention conducted in Fresno, CA, used the same WIM technology [16]. The Fresno research estimated the relative abundance using a Non-Parametric Bootstrapping (NPB) method. This approach finds the mean of an unknown distribution [44], letting the data "speak by itself". Later they estimated 95% confidence intervals from "the 2.5% and 97.5% for all bootstrap[ed]" mean number of females per trap. All of Fresno's TAs had statistically identical peak reductions (T1 = 98.9 (98.1, 99.4)%, T2 = 95 (92, 97)%, T3 = 95 (92, 97)%) to Houston's peak reduction (week 36, 97 (94, 99)%).

Finally, the largest observed reduction in the Harris County intervention went from 0% to 47%, to 82%. We can explain this reduction rate using ecological theory. The size of a population over time ($N_t$) is the result of two processes, recruitment (birth and immigration), and dismissal (death and emigration). In the context of mosquito control, we can say that "chemical pesticides" work mainly by quickly increasing deaths at time $t$, but do not immediately reduce

subsequent births (do not reduce recruitment at *t*+1). This has been empirically observed when the number of adult mosquitoes quickly rebound after an adulticide application. On the other hand, methods like IIT (and SIT) do not increase deaths at *t*, instead they reduced births at *t*+1. For simplicity, assume that migration (immigration and emigration) is non-existent. We set the daily mortality rate to 0.29 [45], the recruitment rate to 0.15 (i.e., 85% suppression by the intervention), and set a starting population ($N_{t0}$) at 10,000. After running the recurrent equation for 14 days, we obtained a reduction of 64% after 7 days, and a reduction of 87% after 14 days, results that are remarkably close to the observed reductions in weeks 29, and 30. Therefore, it is in the realm of possibilities to obtain the observed reductions using an insecticidal approach that only attacks a population's birth rate. A more accurate estimation of population reduction could be achieved with this model, but unfortunately it is currently not possible to accurately estimate the recruitment of field *Ae. aegypti* without conducting large MRR experiments that would have to be conducted during the vector reduction intervention.

Concerning the interaction of the two species. Treatment had a negative influence on the abundance of *Ae. aegypti* in all weeks (Table 1), and it had a (relatively smaller) positive influence on *Ae. albopictus* in all weeks except weeks 36 and 38 (Table 2). Because *Ae. aegypti* was the target of the WIMs, the removal of *Ae. aegypti* may have increased the abundance of *Ae. albopictus*. However, this observation should be view with reservations given the length of our sampling (we likely missed the expansion phase of the population) and the lack of a trend as the number of *Ae. aegypti* was reduced from the TA, however, it is also possible that the TA reached its *Ae. albopictus* carrying capacity after week 30, so further removals of a competing species would not increase its numbers. The observed increase in *Ae. albopictus* appears to agree with the extent to which both species occupy the same larval site. In Florida both species were commonly found together [46] in ovitraps placed by researchers, but in larval surveys in urban areas, both species were discovered to coexist in small percentages [47–49].

Overall, it appears that the releases of WIMs have a marked effect on the *Ae. aegypti* populations where they are released. However, how large, and how quickly the reductions are driven only by the WIMs releases will require more robust experimental designs, which will account for variables that affect the mosquito populations, the trapping, and probably the population aggregation; It is of note that even with the marked decrease in the TA, we continuously found *Ae. aegypti* females during the entire surveillance period. Also, evaluating the impact of WIMs as part of an integrated vector control strategy, and the duration of the effect after ending releases—we monitored for only two weeks after discontinuation of releases—will require further experimentation. At this time, regulation prevents the extensive use of WIMs, which is understandable given the novelty, and the lack of a regulatory framework (at local and federal level) for an insecticidal substance where the "active ingredient" is a live bacterium and the "insecticide formulation" is made of two biological entities. Undeniably, the possibility of having the additional means, together with other complementary tools [50], to eradicate *Ae. aegypti* from large areas of the world is encouraging.

## Supporting information

**S1 File. Dispersion index estimation by week for *Ae. aegypti* and *Ae. albopictus* for both areas.**
(DOCX)

**S2 File. Mosquito abundance estimation program (R markup report) and raw data.** The \*. zip file contains the mosquito abundance estimation program using an N-Mixture model. All the required files (excepting, R, RStudio, and the libraries not in R "base") are included. The analysis is provided as a self-compiling HTML document using the "knitr" library and R-Studio's capabilities. The report is configured to run on a push of button *if the required libraries*

*are installed*. To run the analysis, and create the HTML document, open the "supp_n_mix_abundance_estimation.Rmd" file and click the "Knit" button on RStudio's toolbar. Required libraries: "knitr", "kableExtra", "jagsUI", "sqldf", "ggplot2", "reshape2", "gridExtra".
(ZIP)

**S3 File. Weather and climate data used in Fig 1.** The file contains two spreadsheets ("temperature" and "rainfall") with the daily data used to create Fig 1.
(XLSX)

**S1 Fig. Post-predictive check plots for all weeks, both areas, and both species.** Blue histograms are the best fitted model; Red histograms are the trapping data. A properly fitted model will cover most of the trapping data histogram.
(TIF)

## Acknowledgments

Caroline Weldon, Program Manager of the Western Gulf Center of Excellence for Vector-Borne Diseases (University of Texas Medical Branch); Personnel of Mosquito and Vector Control Division (Harris County Public Health); To the participating homeowners for allowing access to their properties;

**Disclaimer**

The findings and conclusions in this report are those of the authors and do not necessarily represent the official position of CDC.

## Author Contributions

**Conceptualization:** Saul Lozano.

**Data curation:** Saul Lozano, Kevin Pritts.

**Formal analysis:** Saul Lozano.

**Funding acquisition:** Roxanne Connelly.

**Investigation:** Saul Lozano, Kevin Pritts, Chris Fredregill.

**Methodology:** Saul Lozano.

**Project administration:** Dagne Duguma, Chris Fredregill, Roxanne Connelly.

**Resources:** Saul Lozano, Chris Fredregill.

**Software:** Saul Lozano.

**Supervision:** Dagne Duguma, Roxanne Connelly.

**Validation:** Dagne Duguma.

**Visualization:** Saul Lozano.

**Writing – original draft:** Saul Lozano.

**Writing – review & editing:** Saul Lozano, Kevin Pritts, Dagne Duguma, Chris Fredregill, Roxanne Connelly.

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
