## [Decision Letter · Decision Letter 0]

12 Oct 2021

Dear Lozano,

Thank you very much for submitting your manuscript "Independent evaluation of Wolbachia infected male mosquito releases for control of Aedes aegypti in Harris County, Texas, using a Bayesian abundance estimator" for consideration at PLOS Neglected Tropical Diseases. As with all papers reviewed by the journal, your manuscript was reviewed by members of the editorial board and by several independent reviewers. In light of the reviews (below this email), we would like to invite the resubmission of a significantly-revised version that takes into account the reviewers' comments. 

We cannot make any decision about publication until we have seen the revised manuscript and your response to the reviewers' comments. Your revised manuscript is also likely to be sent to reviewers for further evaluation.

Sincerely,

Pattamaporn Kittayapong, Ph.D.

Associate Editor

Amy Morrison, Ph.D.

Deputy Editor

Reviewer's Responses to Questions

**Key Review Criteria Required for Acceptance?**

**Methods**

-Are the objectives of the study clearly articulated with a clear testable hypothesis stated?

-Is the study design appropriate to address the stated objectives?

-Is the population clearly described and appropriate for the hypothesis being tested?

-Is the sample size sufficient to ensure adequate power to address the hypothesis being tested?

-Were correct statistical analysis used to support conclusions?

-Are there concerns about ethical or regulatory requirements being met?

Reviewer #1: The methodology described seems appropriate, with an open discussion on study limitations.

Reviewer #2: Objectives: Not stated. Line 89: The goal of the project was to independently evaluate the efficacy of the WIM release and examine the effects on Aaeg populations. Albo tracking seems to be an afterthought, which was then used as the H1 but neither appear to have been used as a rationale for decision-making again.

Subsection 1) Mosquitoes:

I appreciate that this part of the study was the responsibility of MosquitoMate Inc. however, just because the authors might not have been involved does not mean that it can be glossed over (lines 106-108). Scientific studies are supposed to be reproducible. If this is not possible directly then provide references or the relevant information that will allow the reader to understand the basis of the animals being released into the treatment area. One would assume this to be ref [25]?

Lines 110-112 would read better in a table or include this with a results table of observed data if you want to limit the number of tables. As it currently reads, it's just a list of large numbers that doesn't mean anything as there is no context. Why were these numbers selected? Although it was not the authors decision (since MosquitoMate Inc. was responsible), the reader needs to understand the rationale behind it.

Subsection 2) Monitoring:

Lines 239-241 of the results section should be moved here as they describe the size of both areas.

Line 130: How are mosquitoes "processed"? Do you count them individually? How did you treat other insects/arthropods captured? Numbers? Others mosquito species? How were these identified? References? Did you get any males or just females? All of this information needs to be given to the reader.

Line 134: What is meant by successful? Surely all of the traps would have been checked for correct operation so successful = that they are catching mosquitoes? If experimental interference/malfunction was encountered e.g. dry ice melt or missing components prior to the next trap visit then explain how this was resolved/mitigated.

Line 135: Trapping time is confusing? If all traps were allowed to run for 48-hrs with a change out at 24-hrs, why are you recording mean trapping times and differences between them in between?

Subsection 3) Statistical Methods:

Line 155: It's not that traps do poorly, it's that the trap bait (e.g. CO2) cannot compete as effectively with human CO2 as an attractant to the mosquito

Subsection 4) N-mixture model:

This subsection is not really needed since it's still a statistical method and line 176 repeats line 156. Additionally, lines 177-187 are not methods. It's introduction (see general comments re: intro rework). Methods are for what you did with this data and why, not what somebody else did several years ago.

Line 194: what is "occasion j"? Should it not have two parameters, time t and site s? 

Lines 202-203: this description is rather convoluted. Is it not just the unobserved population? Generally, the descriptions of trapped/trapping/caught = observed and describe the data more appropriately.

Lines 211-212: why was the mosquito count data expected to be aggregated? Explain.

Line 213: Explain why lambda was assumed to follow a gamma distribution.

Line 215: Later? Following what event? Explain the rationale for selecting Taylor's equation.

Line 217: Trapping error is misleading. Unoccupied traps are not trapping errors (which implies a physical issue with the trap) but they are source of sampling error that biases the statistical model. It would probably make more sense to introduce zero-inflation earlier as a common variable of abundance surveys.

Lines 222-224: DI has limitations but these are not detailed nor how they affect this data? 

It would be useful to compile a table of the processes, model distributions, parameters and what each is used for/represents for ease of reference and visualization, as this subsection is requires clarity.

Subsection 5) Testing the N-mixture predictive power:

It is unclear why this exercise is needed if WIMs in this study were marked (since we, the reader, have not been provided with details of the MosquitoMate Inc. release. If they were not marked then:

Line 230: numerically tested needs further explanation re: methodology.

Reviewer #3: The study design was clear and appropriate to accomplish the objectives of the study, with treatment and control sites chosen for release and regular mosquito sampling. The study was conducted in Texas, which has an abundant Ae. aegypti population with demonstrated spread of Zika. It seems a reasonable location to test new control measures, particularly given the potential for compensatory increases in Ae. albopictus, which might be more concerning in an area where DENV is endemic. My primary concern with the Methods – and the paper as a whole – is that the analyses conducted do not integrate the important covariates. The primary objective of comparing mosquito abundance between the treated and untreated locations is done by comparing the mosquito abundance estimates generated from separate N-mixture models. What’s more, this is done on a week-by-week basis, meaning that all statements regarding trends and temporal sequences are based on eyeballing graphs, not on statistical inference. Even if the N-mixture model is so limited that it can only estimate the mosquito abundance and its 95% CI, the negative binomial should have been able to incorporate terms for treatment and week. This becomes especially important for the analyses of Ae. albopictus, which seem to rely entirely on eyeballing the correspondence between the climatic graphs and the Ae. albopictus graphs/estimates.

**Results**

-Does the analysis presented match the analysis plan?

-Are the results clearly and completely presented?

-Are the figures (Tables, Images) of sufficient quality for clarity?

Reviewer #1: The results presented are clear, and match the analysis plan.

Reviewer #2: Experimental Results:

Line 245: what is mean trapping time?

Line 252: what does successful trapping events mean?

Also, it would be more useful if this data was presented in a table for ease of comparison.

Lines 254-256: how did the missing data points affect the overall data?

Model fit and predictive power:

Lines 258-263: These are all statements without reasoning. Rework to link as rationale that supports fit. As it currently reads, "appropriately describes" - based on what? "overlapping histograms" - but to what extent? how much overlap is allowed?, "PPCs" - what are these? are they literally us just looking at the histograms? (Supplementary file 2? If so, then list so we can refer to it), and " DICs" - how do these describe the data other than *? what do the numbers in the table tell us?. More explanation and interpretation required here.

Line 271: insert (n=500) after each color

Line 272: (insert (n=2000) after blue. Why did you only use data from 3/5 releases? Rationale. Without this it appears like cherry-picking.

Lines 272-278: Data presentation and interpretation. Put this in a table. The goal of this exercise is unclear given the current description since there is no direct contrast with how the N-mixture model is superior to the MRR method.

Aaeg rel. and abs. abundance

All of these results would read better if presented in a table then refer to accordingly.

Line 283: estimated relative abundance?

Line 285: what is meant by non-statistical? Statistically insignificant? A change in abundance (an actual observation) that was not statistically different?

Line 289: isn't the "error bar" actually the range of the observed data i.e. the variation?

Lines 289-291: It feels like the data is being interpreted purely statistically rather than biologically. Were any ecological observations made at the time of collections? Did you speak with residents? What about the co-authors who work for the MCVD and their experience? What about comparing this data with the data collected in previous years (I assume MCVD conduct annual surveillance but if not, there was data from June-August 2018 (ref 25) that you might be able to work with. Also, Aedes are great fliers. Have you considered a wind-borne dispersal event using meteorological data? Were there other insects in these traps that might also help explain what happened?

Albo rel. abundance

Again, present results in table and refer accordingly.

Line 318: Albo might not be the intended target of the intervention but apparently it is the H1 of your hypothesis (line 94).

Lines 321-322: Albo has a wider range of observed data in the UA rather than "uncertainty", which sounds like an unknown when it actually isn't. What you are describing is "greater variability".

Line 329: Abundance is driven by reproduction and, given that mosquito larval habitats are aquatic, this is pretty standard. A better way to illustrate this point, however, would be to overlay Figs 5 and 2 as currently, this pattern is not obvious. Even by viewing them both alongside each other.

Line 338: three cold days are probably not going to have much of an impact on the growth rate of the population. It might delay it rather than reduce it.

Line 340: is this suggestion as a result of a conversation with someone e.g. MCVD, or the literature? If the former then state as such (Person X, personal communication).

Line 342-343: I assume you are referring to an increase in abundance but this is not stated. Did the statistically significant increase *definitely* result from the rain? In the same week in the UA, 3.4cm of precipitation did *not* cause a statistically significant increase in abundance (lines 332-333) and this amount was 2x that of the TA (1.6cm) indicating that there are other factors/variables contributing to Albo population dynamics. As the lifecycle is ~8-10d, this is a very tight window for such a increase unless it refers to the very beginning of week 31 and the very end of week 32. I would refer back to the 2018 data to see if there is anything in there that might help explain the 2019 patterns as well as the current observations.

Line 353: typo? Albo not Aaeg? Albo is pretty hardy. Unlikely that the temperature has caused population decreases of this magnitude in such a short space of time. What is more likely, given the circumstances, is wash out of larval habitats, due to Imelda. This is something that occurs regularly in certain islands of the southwestern Pacific as a form of vector control.

Spatial Aggregation

Table 2 is useful but when discussing spatial data it is helpful to have visual representation, in particular Aaeg in the TA to better evaluate the fluctuating DI between weeks. Additionally, It would be nice to see a detailed map of the treated area with trap placements and total catch numbers/trap to identify/eliminate any patterns/observations.

Lines 359 and 361: variability rather than uncertainty.

Line 362: were the WIMs marked as part of MosquitoMate Inc.'s releases? I realize that the authors had no control over this aspect of the study but fluorescent dusting would have been a simple procedure that could have provided physical validation of aggregation.

Line 363: but 1,417,000 mosquitoes were released into 18.6 ha? Is this is one of the limitations of using DI, which you alluded to (lines 223-224) but not in detail? If you expected this to occur, were you forced to use this due to lack of an alternative model/measurement that could have been used?

Line 367: TA, Albo: 3rd and 6th rows are to 3 dec. points. Consistency.

Line 369: Description is square brackets but these don't appear anywhere in the table. 

Figures

1) This map is limited in the information that it provides. I would personally prefer to see greater detail inside the TA and UA of trap placements and the total number of mosquitoes/trap to visualize whether there are any patterns.

2) While this has useful info on it, the y-axis units are too wide to read sufficiently making it awkward to determine values for interpretation of results. Is it possible to include units of 5? Move the "Storm landfall" text away from the LT average black line, which is currently obscuring it.

Line 677: What is meant by expected?

Line 686: Reference?

3) What does this data actually describe? i.e. what is the title of this figure? Lines 258-259 state that fig. 3 "appropriately described the trapping data", and I assume this is based on the fact that the histograms overlay better for the N-mixture model than the neg. binomial? However, what does this actually mean in terms of the what you were investigating i.e. whether WIMs caused Albo populations to increase, which you were measuring by N-mixture models? Either way, this figure needs a title.

Line 690: "... model fitted models". Typo? Statistical fitted models? 

Line 691: "... only even numbered weeks are presented".

Line 692: Define PPCs in full to remind the reader what the measurement is and why it is being applied. Also this information (PPCs .... (lines 692-693) seems more appropriate in the main text)).

4) The title of this figure currently, appears to be, the y-axis, instead of the relative abundance of Aaeg as a function of mean Aaeg F/trap ha-0.1 (lambda) over time.

Line 697: It would be useful to remind the reader how these data points have been calculated i.e. quantiles and refer us to the relevant section in the methodology.

5) Similar to 4) the title of this figure is also the y-axis, instead of the absolute abundance of Aaeg as a function of total number of Aaeg F ha-0.1 (N) over time.

6) Ditto 4) and 5) ... relative abundance of Albo as a function of mean albo F/trap ha-0.1 (lambda) over time. Also useful if dates could be inserted somewhere near the x-axis to interpret experimental observations alongside releases/biologically relevant information/temporal data.

Reviewer #3: Several of the figures, particularly 1, 4 and 6, brought significant clarity to the Results. The order of the Results text was confusing, though, with ‘Experimental Results’ first, but then the abundance results actually not presented in that section, despite those being the entire point of the experiment. My only other major comment about the Results regards the text, “Notwithstanding the PPC results” (line 260), which I don’t see discussed anywhere. The figure in the Supplement is difficult to interpret. Is there something concerning here that calls the other results into question?

**Conclusions**

-Are the conclusions supported by the data presented?

-Are the limitations of analysis clearly described?

-Do the authors discuss how these data can be helpful to advance our understanding of the topic under study?

-Is public health relevance addressed?

Reviewer #1: Overall conclusions are in line with the methodology used, the study limitations and discussion of data.

Reviewer #2: The rationale for WIM releases is briefly mentioned at the introduction but not its impact on human health in the event of its success as an alternative mosquito control intervention.

Conclusions and limitations are supported but inadequately explained. Greater details are required to guide the reader through this research without us having to keep referring back through the paperwork to remind ourselves what was or wasn't done for whatever reason. 

The discussion is not structured. It doesn't remind us of the study's objectives and hypothesis or what is being tested and whether or not the results proved/disproved it. 

Lines 377 and 396: improving the N-mixture model. Meterological data was collected. Why were synthetic datasets not created for LST and cumulative precipitation covariates to simulate a temporal profile?

Line 391: reference?

Lines 397-406: Did you compare this to the previous year's abundance data (2018) for which you had information (ref 25)? What about your coauthors who work for MVCD?

Line 407: This sentence is a statement with no interpretation. Meaning what?

Lines 409-411: refer us to relevant figures.

Lines 419-420: do the MVCD carry out regular surveillance? Are there no other data to make comparisons with? Do the MVCD coauthors have any input here?

Line 422: "safe to assume". This is not a very scientific description.

Lines 424-442: Why is this relevant? Aside from comparing WIM releases, this paragraph abruptly introduces males (line 439), migration rates (line 440), and urbanization (line 442) as statements without rationale. It doesn't place it into the context of this study. 

Line 486: it is definitely within "the realm of possibility" to attack the birth rate since Aedes is primarily controlled though LSM. What is the point here?

Lines 497-500: Might want to rework these lines given Steve's latest publication. See Dobson (2021). J. Med. Ent 58(5), 1980-6

Line 500: It is not feasible to speak in terms of eradicating Aaeg. You can eliminate certain populations, yes, but eradicate, no.

Line 501: New World screwworm

Line 502: ... from the United States, is encouraging.

It remains present in certain Caribbean and South American countries.

Reviewer #3: I find the main conclusion – that the intervention worked – generally compelling despite the limitations of the modeling noted above. The differences shown in Figure 4 are so large, that a more efficient modeling approach would do little to change the conclusion, I imagine. However, I am not at all convinced by the conclusions about the effects on Ae. albopictus. Those results (Figure 6) are too close, and the explanations about climatologic variables need statistical analysis to back them up. Compensatory increases in Ae. albopictus are a serious side effect and have to be critically considered.

**Editorial and Data Presentation Modifications?**

Reviewer #1: (No Response)

Reviewer #2: This maybe addressed at the formatting stage but subsection headings would benefit from bold or italic font.

Bibliography

Line 517: Prevention CfDCa. Doesn't read so well in current format. CDC.

Line 576: MosquitoMate Inc. October. Report No.? Available from where???

Reviewer #3: 1. The full presentation of average number of mosquitoes per trap and total number per hectare-0.1, and inclusion of a figure for both, feels unnecessary, given that they show the same results.

2. I wasn’t aware of this issue before reading this paper, and so had to go do some research on it, but it appears that overdispersion, a distribution with higher than expected variance, has apparently been redefined to some extent in the ecological literature to be conflated with aggregation, the clustering together of population subsets. That seems to be the definition used here, as implied on lines 218-219. However, the kind of overdispersion handled by a negative binomial model is the statistical kind. I am only familiar with it in the Poisson/negative binomial context and cannot judge the appropriateness of this conflation with aggregation in the N-mixture model. I will note, however, that the spatial aggregation results (page 22) don’t really enter into the Discussion/Conclusions in any meaningful way, so the emphasis it receives doesn’t seem justified at present.

3. Lines 96-97. I am not sure what the data from FL is supposed to tell me here.

4. Lines 115-116. On what basis were the sites selected?

5. Line 117. Can you relate this distance to a mosquito’s range to contextualize it for readers?

6. Lines 167-174. Are the parentheses in the correct places here? Order of operations would dictate that TA/UA is multiplied by 100 FIRST, before it’s subtracted from 1. The percent reduction would be (1-(TA/UA))x100.

7. Line 272. Is there a reason all 5 weren’t used? Does the method work equally well in the other two?

8. Lines 274-277. All were underestimates with large confidence intervals. This wasn’t discussed in relation to the study results. How would you anticipate this could be impacting the results?

9. Line 309. What are these referring to?

10. Line 338. Why is the minimum temperature here (18.9) different than the min. of the three days it’s pulled from (line 337)? Typo?

**Summary and General Comments**

Reviewer #1: In this manuscript, Lozano and colleagues perform a “third-party” evaluation of the effect of field releases of Wolbachia-infected males by MosquitoMate, as part of their IIT approach. This study, conducted in the metropolitan area of Houston, TX found, through a Bayesian hierarchical estimator approach, reductions of 96% in Ae. aegypti population after around 6 weeks of field releases. The manuscript is well-written, the data is clear, and the methodology is adequate. My main suggestion (and it is only a suggestion) is to move all the Ae. albopictus dataset as supplementary material, as the original hypothesis/motivation for such comparison was not strongly supported by the results presented. It also does not represent the intended goal of MosquitoMate’s IIT approach, while distracting the reader from the main objective, which is to see reductions in Ae. aegypti population size. That said, the study design has limitations that were, in my opinion, reasonably addressed by the authors. I have only a few small suggestions that should be corrected.

Comments shown in order of appearance:

Line (L) 39-40: I suggest rephrasing the sentence to use the correct terminology and inform that the species transmit many pathogens with the potential to cause diseases in humans, instead of carrying diseases.

L. 44: Please complete the sentence by describing that CI happens in crosses between Wolbachia-infected male x uninfected female.

L. 46: “The bacteria are passed from laboratory raised males to field females via copulation…”

This is incorrect. Cytoplasmic incompatibility is induced by modifications to the sperm in Wolbachia-infected males causing a break down and asynchrony in timing of developmental events between male and female gamete pronuclei. No Wolbachia is transferred via mating. 

L. 47:”…the males are infected with Wolbachia and later released in large numbers to outcompete wild males.”

As it currently reads, the text reads as males are routinely transinfected with Wolbachia in the lab before field releases. Please rephrase to reflect that this is a one-time-only process that generates a stably-transinfected colony.

My suggestion would be something along the lines of: Wolbachia-infected males are released in large numbers (inundative release) as to outcompete wild-type Wolbachia-free males.

L. 68-69: Sentence could be shortened given the prior statement.

L. 94: reference 18. I find this reference not entirely appropriate for this context. I suggest the following as alternatives:

https://pubmed.ncbi.nlm.nih.gov/1791461/

https://pubmed.ncbi.nlm.nih.gov/7650719/

L.389-396 // 418-423: Good examples of good discussion of results/addressing limitations.

Reviewer #2: This study is addresses a common ecological issue, that of estimating population abundance. However, the text is confusing due to a lack of structure and the story is lost in verbosity and numbers. This work would greatly benefit if a lot of methods and results were placed in tables for ease of reference (and is also more visually appealing). Additionally, these sections would flow better if they were written in the same order as their subsections.

Abstract:

Pretty light on methodology and results. I don't expect to read about other studies here (lines 34-35 and 36-37). The abstract is your headliner. It's supposed to make your work stand out and grab the reader's attention. This, does not.

Introduction:

Line 64. This is incorrect. Dengvaxia is licensed and approved for use. It may not be the primary method of control but it is incorrect to say that there is no working vaccine. Generally, this section needs reworking to introduce the concept of why estimating abundance is so challenging and how the gold standard (MRR) is limited necessitating statistical models but how they themselves are prone to bias if they are unable to simultaneously model true and false sources of variation in a manner that accurately represents underlying ecological mechanisms and observation errors.

Reviewer #3: Lozano et al. performed an independent investigation of a commercial Ae. aegypti control intervention that avoids building concerns regarding insecticide resistance using an Incompatible Insect Technique. Leveraging treated and untreated areas with Ae. aegypti released by the manufacturer, the investigators collected mosquitoes at regular intervals. They used a novel statistical technique to account for underestimates of mosquitoes in traps and ultimately showed drastic decreases in Ae. aegypti in the treated zone. They argue that apparent increases in Ae. albopictus in the treated area are due to climatic variation and not due to the intervention. Overall, the study was well-conducted and compellingly shows that that the Incompatible Insect Technique is successful in reducing Ae. aegypti abundance. However, limitations of the statistical methods used leave open the question of whether the Ae. albopictus population responded with increased abundance, which is an important public health consideration.

PLOS authors have the option to publish the peer review history of their article (what does this mean?). If published, this will include your full peer review and any attached files.

Reviewer #1: Yes: Heverton Leandro Carneiro Dutra

Reviewer #2: No

Reviewer #3: No
---

## [Decision Letter · Decision Letter 1]

20 Jul 2022

Dear Lozano,

Thank you very much for submitting your manuscript "Independent evaluation of Wolbachia infected male mosquito releases for control of Aedes aegypti in Harris County, Texas, using a Bayesian abundance estimator" for consideration at PLOS Neglected Tropical Diseases. As with all papers reviewed by the journal, your manuscript was reviewed by members of the editorial board and by several independent reviewers. The reviewers appreciated the attention to an important topic. Based on the reviews, we are likely to accept this manuscript for publication, providing that you modify the manuscript according to the review recommendations. 

Sincerely,

Amy C Morrison, PhD

Section Editor

Amy Morrison

Section Editor

There was consensus among the editorial board that this manuscript should have a path to publication, and ask that your revised manuscript carefully address the remaining comments from Reviewer #3. In addition, below are some observations from the Section Editor that I suggest the authors consider. No need to provide a point by point response to the editor comments, but hope they may influence the next revision.

1. The manuscript presents data from an independent evaluation if a Wolbachia-based population suppression strategy in the city of Houston, TX. From this data, there is compelling evidence that the strategy was effective during the period of evaluation, even though the study methodology had numerous limitations, including but not limited to small sample size (only 26-28 BG traps each in an untreated an treated area, no replicates (one treated and untreated area) and for me no significant comparison of the two areas prior to implementing the intervention.

2. The authors use a relatively novel statistical approach to analyze the data which seems reasonable to me. It is clear the authors have a strong grasp of the statistics and are able to discuss the advantages and disadvantages over other methods such as negative bionomial regression. As far as I can discern the trap data is pretty convincing anyway you do the analysis. The argument that the their method accounts for trapping biases was not all that convincing to me especially since they used a single mark release capture study to "parameterize" (this was my non-statistical interpretation).

3. Those of us who sample Aedes aegypti, no that there are always a lot of houses with zero trap totals (or really don't have Ae. aegypti), work I've don't personally shows that distributions don't have much spatial structure beyond an individuals household, but the distribution of infested houses change overtime. Our conclusion has always been you need large sample sizes. Again this study was not outstanding in size but large enough in my view to measure the impact of the intervention.

4. I have reviewed and edited other articles using this technology and the companies unwillingness to provide methodological details is always frustrating, but beyond the authors control.

5. Reviewer #2 clearly had some important insights, but the interaction became contentious and unproductive. I do think the manuscript improved in in response Reviewer #2s comments.

6. Other optional suggestions from the editor beyond addressing Reviewer #3' queries.

- I would recommend that in addition to that, that the authors attempt to reduce the results and discussions sections by at least 50%.

- I would focus on what was observed, specifically for Aedes aegypti and at least a paragraph on if cities in the US should consider using this technology

- Consider eliminating the discussion of the impact of rainfall and temperature. As you point out, the issues could not be addressed statistically. You might indicate how large a study would be required to do so adequately and stop there, with one sentence indicating that clearly these factors helped drive some of the temporal variation you saw.

- The discussion of the larval habitats for aegypti versus albopictus, was interesting, but a better designed study would be needed to address this question.

- A brief discussion of the merits of this statistical approach over others

- You did a good job discussion the design limitations but provide a concise summary and let it rest.

- I make those suggestions, because I felt the manuscript rambled endlessly and obscured a clear and important finding, 95% reduction in adult female aegypti densities.

-You might consider moving more of the text to supplementary information and keeping the main body of the manuscript much simpler.

Again, the above paragraph are suggestions.

Amy

Below are the reviewer comments.

Reviewer's Responses to Questions

**Key Review Criteria Required for Acceptance?**

**Methods**

-Are the objectives of the study clearly articulated with a clear testable hypothesis stated?

-Is the study design appropriate to address the stated objectives?

-Is the population clearly described and appropriate for the hypothesis being tested?

-Is the sample size sufficient to ensure adequate power to address the hypothesis being tested?

-Were correct statistical analysis used to support conclusions?

-Are there concerns about ethical or regulatory requirements being met?

Reviewer #2: (No Response)

Reviewer #3: I believe the methods for determining whether the Ae. aegypti population decreased were adequate. The additional analyses conducted in response to Reviewer #3 could be placed in the Supplement as sensitivity analyses. With the most recent changes, validation of the N-mixture method is now also adequate.

However, concerns remain regarding the Ae. albopictus analysis. See comments on Results.

**Results**

-Does the analysis presented match the analysis plan?

-Are the results clearly and completely presented?

-Are the figures (Tables, Images) of sufficient quality for clarity?

Reviewer #2: (No Response)

Reviewer #3: Though not intuitively organized, as pointed out by Reviewer #2, the Results are largely sound and most issues have been resolved. 

Starting with the statement on line 324, "Interestingly, the week-to-week variation appeared to be related to rainfall events (Fig. 1)," and continuing through the rest of the section to line 350, the authors spend 400 words describing rainfall and temperature fluctuations that may have influenced Ae. albopictus levels instead of the intervention. The authors are most frank in their responses to reviewers about the limitations of the Ae. albopictus analyses. They seem to feel that because they have not attached statistical language to these findings and have occasionally inserted words such as "probably" in front of their suppositions that the inclusion of these "findings" is legitimate. Reviewer #2 explains biological mechanisms that would make it unlikely that some of the relationships the authors claim as findings actually explain the data observed, yet these comments have not been adequately addressed. More fundamentally, I simply do not view these as "results." The authors' rationale that they cannot perform any statistical analysis to associate the weather patterns with the Ae. albopictus levels because of the low number of weeks sampled does not open the door for them to assert whatever they want -- a range of relationships in precipitation and temperature levels, with varying number of weeks between the weather event and the supposed responding mosquito bloom.

Additionally, the spatial aggregation results do not figure into the Discussion and Conclusions. As such, I believe they could be removed or relegated to the Supplement, reducing the length and complexity of the paper.

**Conclusions**

-Are the conclusions supported by the data presented?

-Are the limitations of analysis clearly described?

-Do the authors discuss how these data can be helpful to advance our understanding of the topic under study?

-Is public health relevance addressed?

Reviewer #2: (No Response)

Reviewer #3: In line with the comments on the Results, I think the conclusions relating to weather in the Discussion need to be significantly scaled back and be shaded with substantially more caution. The statement added about skepticism regarding the possibility of the intervention increasing Ae. albopictus levels goes in the wrong direction -- there is some suggestion in this data that this occurred, and nothing the authors have presented provides a compelling reason to disregard that signal. The small number of weeks presented is not a reason to disregard it when there could be harm done by ignoring it.

**Editorial and Data Presentation Modifications?**

Reviewer #2: (No Response)

Reviewer #3: (No Response)

**Summary and General Comments**

Reviewer #2: (No Response)

Reviewer #3: Several comments from the original reviews were not addressed, indicated as addressed but not actually addressed, or answered in the response to the reviewers but not addressed in the manuscript. The original comments should be re-reviewed to ensure all have been appropriately addressed. In general, if a reviewer is confused about something, it should be clarified. Particularly, those reviewers that appeared to be subject matter experts should be considered as the target audience, and the paper clarified such that it would be understandable to them without additional explanation.

PLOS authors have the option to publish the peer review history of their article (what does this mean?). If published, this will include your full peer review and any attached files.

Reviewer #2: No

Reviewer #3: No

Figure Files:

Data Requirements:

Reproducibility:

References

---

## [Editor Report · Decision Letter 2]

23 Oct 2022

Dear Lozano,

We are pleased to inform you that your manuscript 'Independent evaluation of *Wolbachia* infected male mosquito releases for control of *Aedes aegypti* in Harris County, Texas, using a Bayesian abundance estimator' has been provisionally accepted for publication in PLOS Neglected Tropical Diseases.

Best regards,

Amy C. Morrison, PhD

Section Editor

Amy Morrison

Section Editor

Thanks for addressing the remaining reviewer and editor concerns.

The manuscript is being accepted but during the processing, on line 386 view should be viewed.

---

## [Editor Report · Acceptance letter]

8 Nov 2022

Dear Lozano,

We are delighted to inform you that your manuscript, "Independent evaluation of *Wolbachia* infected male mosquito releases for control of *Aedes aegypti* in Harris County, Texas, using a Bayesian abundance estimator," has been formally accepted for publication in PLOS Neglected Tropical Diseases.

Best regards,

Shaden Kamhawi

co-Editor-in-Chief

Paul Brindley

co-Editor-in-Chief
